# Real World Evidence on the Effectiveness of Nusinersen within the National Program to Treat Spinal Muscular Atrophy in Poland

**DOI:** 10.3390/healthcare11101515

**Published:** 2023-05-22

**Authors:** Dominika Krupa, Marcin Czech, Ewa Chudzyńska, Beata Koń, Anna Kostera-Pruszczyk

**Affiliations:** 1Faculty of Management, University of Warsaw, 02-678 Warsaw, Poland; 2Department of Pharmacoeconomics, Institute of Mother and Child, 01-211 Warsaw, Poland; 3Department of Analysis and Innovation, National Health Fund, 02-528 Warsaw, Poland; 4Department of Neurology, Medical University of Warsaw, 02-097 Warsaw, Poland

**Keywords:** spinal muscular atrophy, real-world evidence, nusinersen, therapy costs, budget impact

## Abstract

Background: Spinal muscular atrophy (SMA) is a debilitating neuromuscular disease resulting in children’s mortality and disability. Nusinersen is available to all SMA patients in Poland since 2019. Aim: To compare mortality or disease progression to mechanical ventilation in two patient cohorts before and after the program’s introduction. Additionally, to describe the patient population treated with nusinersen and costs incurred by the public payer. Methods: We used the National Health Fund (NHF) database to identify patients born in either 2014 or 2019, who received at least two health services with an ICD10 G12 diagnosis. Outcomes were time to event: death or first mechanical ventilation. We identified all benefits received by nusinersen-treated patients, between 1 January 2019 and 31 May 2022. Results: Children with SMA born in 2019 had significantly lower mortality in the first years of their lives than children born in 2014. Approximately 875 patients (all age groups) were treated with nusinersen in the analysis period. The cost of causal drugs in this period amounted to €51.4 million. The cost of healthcare benefits amounted to €14.9 million. Conclusions: The drug program to treat SMA improved patient care in Poland. The NHF database was a reliable source to monitor resource-intensive therapies’ costs, demography, and selected patient outcomes.

## 1. Introduction

Spinal muscular atrophy (SMA) is a debilitating neuromuscular disease, caused by mutations of the Survival Motor Neuron (*SMN1*) gene located on chromosome 5q13 [1]. The mutation results in a deficiency of the SMN protein, which leads to the death of motor neurons in the spinal cord. The *SMN1* gene has a nearly identical copy to the *SMN2* gene. The number of copies of the *SMN2* gene is a strong phenotype modifier, and in most patients is inversely associated with disease severity. The disease is characterized by degeneration of lower motor neurons, which leads to progressive muscle weakness and atrophy. Age of onset of symptoms and highest achieved motor milestones led to the definition of subtypes of the disease [2]: SMA1 is symptomatic in infancy, up to 6 months after birth. Children with SMA1 never achieved the ability to sit unsupported; SMA2 patients become symptomatic between the ages of 6–18 months and could sit but never walked unsupported; patients with SMA3 can walk without support, and many lose ambulation in the course of the disease. SMA4 patients are diagnosed in adulthood and present with limb-girdle weakness. Historically, the median age at reaching the combined endpoint of death, or at least 16 h/day of respiratory support in SMA1, was 13.5 months [3].

The incidence of SMA in Europe was reported as 1 in 3900–16,000 live births [4]. The prevalence of the disease is estimated at 1–2 cases per 100,000 people [2]. With improved access to genetic testing and the introduction of newborn screening programs, accurate epidemiological data become available. The estimated incidence of SMA in Poland is 1:7356 live births [5]. The approximate SMA patient population size has been estimated at 1100 cases [6].

Up until recently, only symptomatic management and palliative care were available for SMA patients. The first disease-modifying agent, nusinersen (Spinraza^®^, Biogen) was granted marketing authorization by the Food and Drug Administration (FDA) in the USA in 2016, and by the European Medicines Agency (EMA) in 2017. As of 1 January 2019, the drug has been made available in Poland through a dedicated drug program to patients of all ages, with all SMA types, irrespective of the symptom severity or *SMN2* copy number [7]. In September 2022, the available treatment catalog was extended to include further entries into this clinical space: onasemnogen abeparvovec (Zolgensma^®^, Novartis) and risdiplam (Evrysdi^®^, Roche).

A national screening program for SMA in newborns was introduced in Poland in April 2021, and gradually implemented across the country. Since March 2022 it includes every newborn (opt-in necessary) [8].

The clinical effectiveness of therapy with nusinersen in our country has been reported before [6]. None of the treated patients were discontinued due to ineffectiveness, and in most cases a significant functional improvement was observed, measured by the Children’s Hospital of Philadelphia Infant Test of Neuromuscular Disorders (CHOP-INTEND) or the Hammersmith Functional Motor (HFMSE) Scale Extended, as appropriate for the patient’s age and baseline function. The therapy was well tolerated and effective, which provides further rationale for introducing the program. We intend to further extend that conclusion by examining the impact of therapy on patients’ overall survival.

The aim of this study is to describe the demographics of the SMA patients treated in Poland, the costs incurred by the public payer, and the effectiveness of the treatment of SMA in clinical practice.

## 2. Materials and Methods

The primary data source for the epidemiology and cost analyses was the claims database of the National Health Fund (pol. Narodowy Fundusz Zdrowia, NFZ), which contains information about all healthcare benefits financed from public sources. Since the NFZ is the sole public payer in Poland, and in the absence of significant private medical insurance, especially in a highly specialized and costly treatment such as SMA therapy, it is reasonable to assume that the database covers the entire population of interest. The NFZ claims the database enables longitudinal patient-level analysis based on database-wide unique identifiers. Therefore, it enables tracking of selected patients across the entire publicly financed healthcare system, including prescription drugs in community pharmacies, medical device use, inpatient care, and specialty outpatient care. Instances where benefits are financed via flat rates, such as primary care, emergency room (including ambulances), or health resort treatments, where assigning a financial figure to a specific benefit granted to a specific patient is not possible, are excluded from analysis.

We analyzed the demographics of SMA patients enrolled in the drug program, from its establishment in January 2019 up until May 2022 (cut-off date). The same period was used to analyze costs associated with the program. The date of first reported administration of the drug was adopted as the date of enrollment, all nusinersen administration dates were reported to the NFZ for each patient. Both cost and demographic data have been augmented with details of the SMA type sourced from a dedicated platform created by the payer for the monitoring of the drug program. It contains additional patient information besides the claim’s records stored in the NFZ.

To analyze the real-world effectiveness of therapy with nusinersen in Poland, we adopted a targeted approach, with the objective to select two groups of patients whose only identifiable differentiating characteristic was access to treatment. We compared two cohorts of patients, children born in 2014 and 2019, where the latter cohort is assumed to have had access to treatment from birth (as they were born after the introduction of the program) and the former cohort who only acquired access at a later age when the program was introduced in Poland. Both cohorts were eligible to receive the treatment with nusinersen upon the program’s introduction. We looked at two major endpoints: overall survival (OS) and time to progression, defined here as either the first reported instance of mechanical ventilation or death. The reasoning for narrowing down the population to the two distinct cohorts was to control for potential confounding factors related to patient characteristics and accessibility of specialized care within the setting of the Polish healthcare system. 

Identification of patients with SMA from distinct cohorts was based on two separate records of ICD10 code G12.X diagnosis in patients’ medical history. The diagnosis had to have been issued in a hospital or by a neurology specialist. The patient was later followed up based on his or her unique identifier, regardless of other concomitant diagnoses or reasons for interaction with the public healthcare system. It is the same approach as published in [5], that reduces the probability of including patients with a single or unsure diagnosis. Retrograde analysis of patients with SMA included in the drug program of patients based on inclusion into the drug program would be insufficient since many of the patients born in 2014 have died or were put on mechanical ventilation before nusinersen was available. Therefore, they would have been excluded from the analysis, inflating the impact of therapy compared to those who are tracked with confirmed diagnoses from birth.

## 3. Results

### 3.1. Effectiveness

We identified 45 patients born in 2014 who have had at least two separate healthcare services recorded within the NFZ database with the ICD10 code G12 and its extensions. Applying a similar methodology, 51 patients born in 2019 were included in the analysis. Up to the data lock date set on 31 May 2022, there have been eight deaths in the 2014 cohort and two in the 2019 cohort. On top of that, 20 patients from the 2014 cohort were administered any duration/day of mechanical ventilation compared to 23 from the 2019 cohort. 

Figure 1 presents the Kaplan Meier curves for mortality within the two groups—patients born in 2014 and patients born in 2019. Out of children born in 2014, 72% of the cohort survived past their fourth birthday. No events were recorded for the children between the ages of four and eight. In the younger cohort, 97% of the cohort reached the three-year mark. The difference in overall survival is statistically significant at *p* < 0.05 (test statistic of the Log-rank test Χ^2^ = 4.9, *p* = 0.03).

While the difference in mortality is significant, when survival analysis is extended to include mechanical ventilation, as presented on Figure 2, the difference in results between the cohorts becomes insignificant (*p* = 0.93, test statistic of the Log-rank test Χ^2^ = 0.01), as presented. 

Median survival for patients in the 2014 cohort was 823 days, or two years and three months. For the 2019 cohort, median survival was not observed as of the cutoff date of day 1382 since birth.

### 3.2. Demography of the Treated Cohort

In the period between January 2019 and May 2022, 875 patients have been included in the drug program and treated with nusinersen. As stated above, the program is dedicated to all patients with genetically confirmed SMA diagnosis; therefore, the described population is assumed to be equivalent to the total known patient population in Poland at the time. 

Out of the total population, 53% were male and 47% were female. The average age was 21.7 years for males and 21.3 for females. Furthermore, 53.7% of male patients and 51.6% of female patients included in the program were under 18 years of age. The differences between the groups were not statistically significant. The detailed split is presented in Table 1.

In 2019, the first year of the program, 442 patients were included in the treatment: 137 patients had SMA type 1, 105 had SMA2, and 196 had SMA3. Four patients started treatment presymptomatically. The size of the treated population by SMA subtype in subsequent years is presented in Figure 3. The share of SMA1 patients declined from 31% in the first year to 22% in the most recent period. The share of treated SMA2 patients was constant throughout the period at 24% of the population. In 2022, a little more than half of the treated population had been diagnosed with SMA3.

As mentioned earlier, in the first year of the program, 442 patients started treatment. In subsequent years, there were 280, 134, and 19 initiations respectively (up to the cutoff date). There have been 15 treatment discontinuations in 2020, 50 in 2021, and 48 in the first five months of 2022. The size of the treated population at the cutoff date was 762. Figure 4 presents the flow of patients within the program since its onset.

### 3.3. Costs

The direct medical costs associated with the treatment of spinal muscular atrophy covered within the public system are the costs of disease-modifying therapy and the costs of healthcare benefits granted to patients. 

Between the introduction of the drug program and reimbursement of the first disease-modifying therapy in Poland in January 2019 and May 2022, the costs of medication amounted to 118 million Euro—13.96 million in the first year, 29.8 million in the second year, 36.86 in the third year, and 37.46 million in the first five months of the fourth year of reimbursement, as presented on Figure 5. The unit price of the product is most likely covered by a confidential agreement between the market authorization holder (MAH) and the Ministry of Health (MoH) [9]. Therefore, while the total cost of therapy is reported in public sources, the exact figure per patient is not known, as the nature of the agreement is not disclosed.

Costs of healthcare benefits were aggregated at the individual patient level, starting with the identification of all unique patients who have received the drug within the drug program, and the selection of all benefits reported within this population. The healthcare benefits were subsequently grouped into the following categories:Outpatient specialist care (excluding psychiatric care and medical rehabilitation)Inpatient care (excluding psychiatric care and medical rehabilitation)Benefits related to the drug program (diagnostics and administration of the drug, excluding the cost of the active substance)Long-term careMedical rehabilitationPalliative and hospice careReimbursement of prescription pharmaceuticals, special foods, and selected medical devices (blood glucose strips and dressings, needles)Reimbursement of medical devices

The total cost of healthcare benefits granted to patients within the same period amounted to 14.86 million euros. The largest cost driver among benefits was long-term care, which cumulatively accounted for 47% of the total cost. The overall cost structure can be characterized as relatively stable. The year 2020 may be treated as an exception due to the unprecedented situation related to the coronavirus pandemic, which resulted in restrictions on the accessibility of healthcare services and potential deviations in reporting benefits. Despite the decrease in the cost of benefits in 2020, attention should be paid to the fact that the population of patients who were offered treatment increased significantly. Figure 6 presents the detailed split of healthcare benefit costs between January 2019 and May 2022.

As previously mentioned, the types of SMA are differentiated by the course of the disease and the severity of symptoms. SMA1 manifests at the earliest stage of a child’s development and is the most severe, while SMA 3−4 have a chronic, milder course. This is also reflected in the cost of caring for patients with specific types of diseases. Although SMA1 patients constitute between 20 to 30% of the treated population within the Polish drug program (depending on the year of analysis), their treatment accounts for nearly 60% of the total cost. A detailed breakdown of disease types and share in total costs is presented in Figure 7.

Long-term care accounts for almost 60% of the cost of care for patients with SMA1. In the case of SMA2, it is 38%, and for SMA3 it is just over 16%. This is related to the more severe course of the disease of these patients, and thus much greater patient needs for this type of service. Regardless of the type of disease, hospital treatment accounts for about a quarter of the total cost. For the group of patients with SMA3, costs related to the drug program are an important cost item, i.e., the costs of diagnosis, monitoring, and administration of the drug. In more severe forms of the disease, SMA types 1 and 2, this item has a decreasing share in the total costs. Patients with SMA3 use medical rehabilitation services to a relatively greater extent. A detailed breakdown of the shares of individual categories in the total costs of patient care is presented in Figure 8.

The average cost of treating a patient within the drug program decreased significantly from the onset of the program, where it amounted to almost 10,711 euros annually, down to 5481 in 2022 (full-year extrapolation from partial data covering Jan–May). Figure 9 shows the average costs in individual categories per patient treated in the drug program. After the first year of the program’s operation in Poland, stabilization of the costs of treating a single patient in all categories is observable. Long-term care benefits remain the largest cost, averaging around three thousand euros per year. The cost of hospital treatment is about two thousand euros per year. The costs of a patient’s participation in the drug program amount to approximately 600 euros, similar to the costs of medical rehabilitation and medical devices. The average costs naturally vary by disease subtype, as healthcare benefits for patients with SMA1 amount to an average of 15,611 euros per year, for SMA2 to 6160 euros, and for patients with SMA3 to 2653 euros per year.

## 4. Discussion

This is the first study to comprehensively describe Polish SMA patients in terms of treatment and care costs. A key strength of the data presented herein is the fact that they pertain to the total population of patients with SMA. Data on Polish patients were analyzed previously [5] and focused on the epidemiology and standards of care for patients with SMA before the initiation of the drug program, which granted access to the first causal treatment to all patients. This work builds on results published there, as we greatly extended the size of the population covered. 

Our results suggest that the measures recently introduced in Poland to screen, diagnose, and treat patients are effective tools for managing this debilitating disease.

Firstly, our results confirm that treatment with nusinersen leads to a significant reduction in the overall mortality of children with SMA in our real-world cohort. It was previously demonstrated in the setting of a prospective, strictly controlled clinical trial [10,11]. Inclusion criteria for the drug program in our country allow to start treatment in patients across the complete SMA severity spectrum, also those with symptoms more advanced as compared with clinical trial inclusion criteria. Overall, out of children born in 2014, 72% of the cohort survived past their fourth birthday. In those born in 2019, 97% of the cohort reached the three-year mark. Interestingly, we found no difference between treated and untreated cohorts in terms of progression to any duration/day of ventilation. This may suggest that advances in therapeutic options and awareness of SMA can extend a patient’s life but are not always successful in postponing the onset of ventilation. It may also be because the drug program requires frequent evaluation by a multidisciplinary team, which may trigger the pro-active introduction of ventilatory support, for fewer than 16 h/day. The data collection method did not allow us to verify the number of hours of mechanical ventilation received. It has also been demonstrated that functional improvement with nusinersen treatment builds up with consecutive doses; thus, possibly longer observation is needed [6,12].

Secondly, we observed a dynamic increase in the number of patients treated within the drug program during the first two years since the reimbursement decision. We hypothesize that the drug program included most older patients with SMA and that most new initiations will be likely due to new diagnoses, including newborns identified by the nationwide screening program. This strengthens the role of rare disease registries in trial- and treatment readiness. Our SMA TREAT NMD Registry facilitated planning treatment [13,14]. The size of the treated population is relatively large and is likely to grow in the future, as current patients will continue therapy and new initiations will occur at a steady pace related to the natural incidence of the disease. This enables policy planners to adequately plan capacity in neurology centers in the country in terms of resources. 

Thirdly, we conclude that caring for patients with spinal muscular atrophy is cost-intensive, compared to the average cost of patient care in Poland, which amounted to 582 EUR in 2021 [15]. It is, therefore, reasonable to monitor and analyze costs to maintain control over them and identify anomalies (e.g., sudden increases) early. The analysis of the costs of care for all patients included in the program until May 2022, as previously presented, suggests that the payer’s expenses per patient were stable after having decreased after the introduction of the drug program. In the period 2019–2021, average NFZ expenses per patient increased by a compound annual growth rate of 3.9% [15]. This may indicate the development of certain standards in dealing with patients on a national scale.

Our study has several limitations. The treated cohort is heterogeneous in terms of age and severity, and while we report on costs by SMA type, we did not analyze the dynamic of costs depending on patient age. In the effectiveness section, we aimed to address the heterogeneity by comparing groups of the same age. Due to the limited scope of information in the claims database, we could not stratify the cohorts by disease severity. Furthermore, the duration of observation is relatively short as compared with the natural history of SMA 2−3, or chronic types. It is possible that some severe SMA1 neonates who were born with the disease died before confirmatory diagnostics. The inability to identify those patients within the database, and their resulting exclusion from analysis, limit the mortality analysis but should have a similar influence on data from 2014 and 2019 (pre-newborn screening), as a result, older cohorts have milder phenotypes resulting in improved survival. For this reason, we chose a recent cohort to provide a comparison to those born before and after wide accessibility of treatment. It is possible that some patients that should have been included in our analysis have been missed. 

Another limitation is related to the selection of cohorts—we compare only two groups of patients, where a broader, more nuanced analysis by subpopulations would be warranted to perform sensitivity analyses on the presented results. The study presented herein is a retrospective analysis of an insurance database, which is covered by strict privacy clauses that prevent the identification of individual patients, limiting the extent of patient characteristics used. We were not able to evaluate indirect costs related to SMA. In addition, despite the publication of the ICD-11 by the WHO [16], Poland has not fully introduced the new classification into the reporting system and clinical practice as of May 2023. Therefore, in our research, we reference the commonly used, yet technically outdated, ICD-10 to present the methods utilized in the research and maintain consistency with previous local studies in the field.

## 5. Conclusions

Poland introduced policy tools that significantly improved the situation of SMA patients in our country—a newborn screening and drug program that grants access to treatment to all patients with a genetic diagnosis of SMA. Extended access to therapy introduced in 2019 successfully reduced patient mortality compared to the situation before the program’s introduction.

As of May 2022, 875 patients were treated with disease-modifying therapy. The treated population is likely to expand in time due to new initiatives and reduced patient mortality. The bulk of the costs of the treatment was associated with the cost of the drug, which is covered by a confidential agreement between MoH and MAH. The costs of healthcare benefits granted to patients were stable on a per patient basis, which suggests the development of standards of care for patients. Our results can be used as a benchmark for resource planning by policymakers in other countries who may consider the introduction of an SMA treatment program and its scope. Further research is warranted to expand on our findings in terms of real-world treatment effectiveness in specific subpopulations and total costs related to SMA treatment, including indirect expenses.

## Figures and Tables

**Figure 1 healthcare-11-01515-f001:**
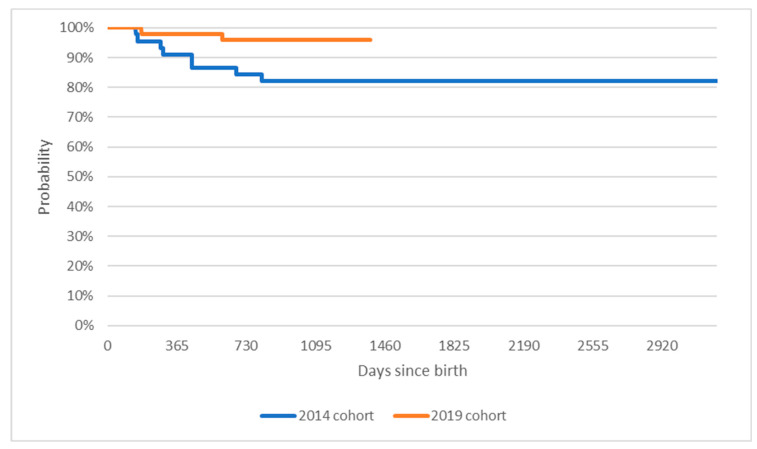
Kaplan-Meier curves comparing time to death of patients born in 2014 or 2019.

**Figure 2 healthcare-11-01515-f002:**
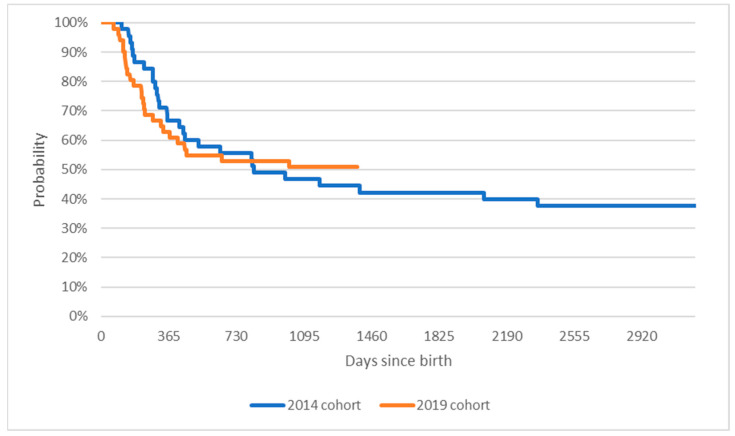
Kaplan-Meier curves comparing time to death or mechanical ventilation of patients born in 2014 or 2019.

**Figure 3 healthcare-11-01515-f003:**
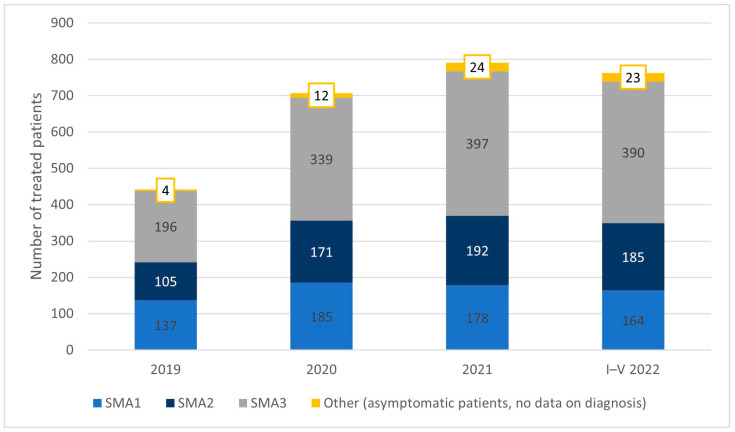
The population of patients treated with nusinersen within the drug program, 2019–May 2022.

**Figure 4 healthcare-11-01515-f004:**
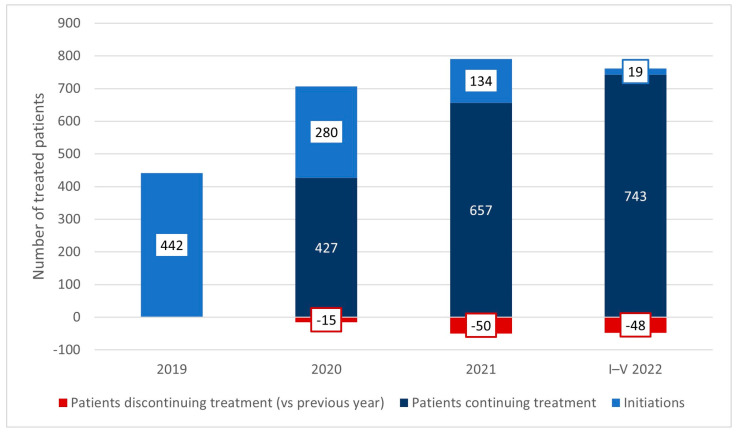
The flow of patients within the drug program, 2019−May 2022.

**Figure 5 healthcare-11-01515-f005:**
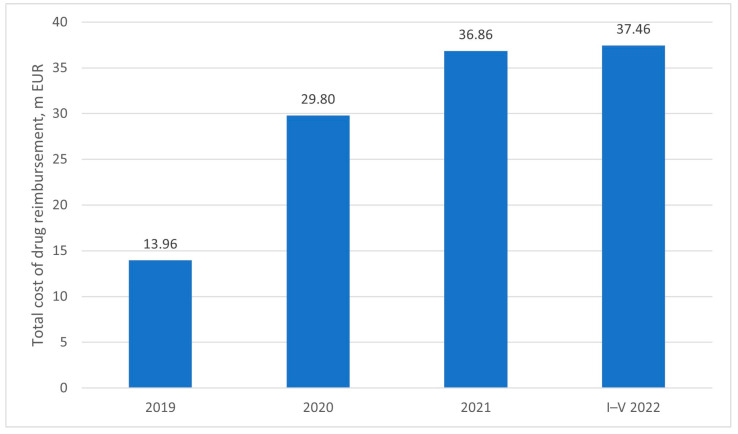
Cost of reimbursement of nusinersen within the drug program.

**Figure 6 healthcare-11-01515-f006:**
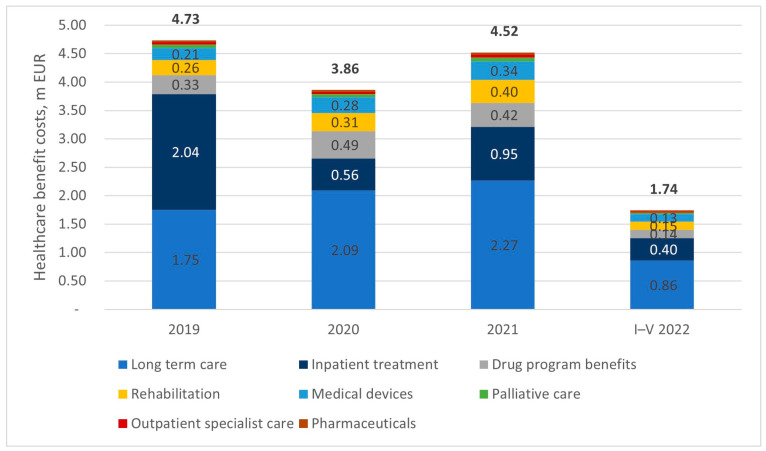
The detailed split of healthcare benefit costs between January 2019 and May 2022.

**Figure 7 healthcare-11-01515-f007:**
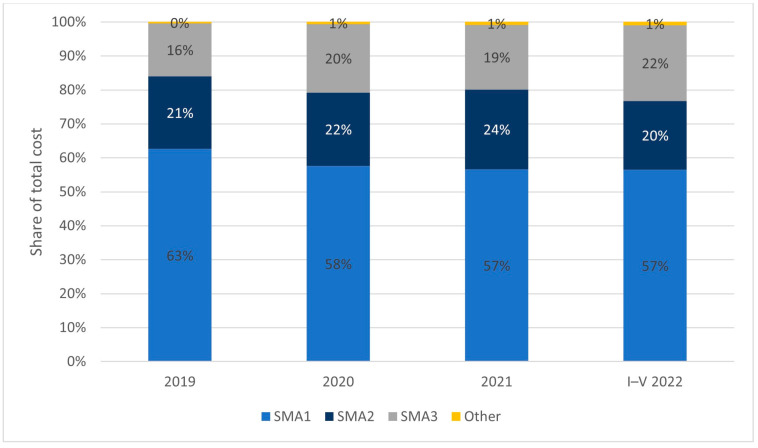
Split of healthcare costs by patients’ subtype of SMA.

**Figure 8 healthcare-11-01515-f008:**
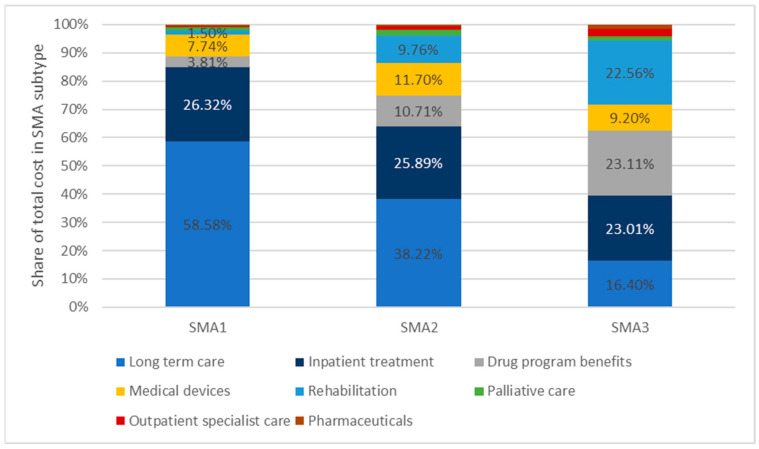
Costs covered by the public payer by healthcare benefit category between January 2019 and May 2022 by subtypes of SMA and types of care.

**Figure 9 healthcare-11-01515-f009:**
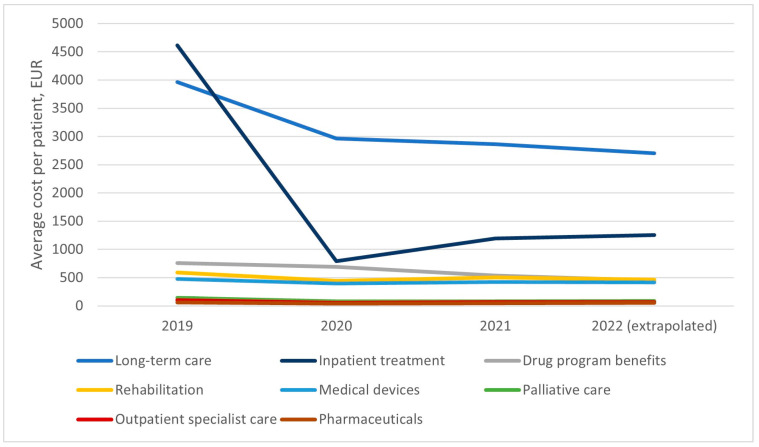
Average healthcare benefit cost per patient treated within the drug program. Note: * Data for 2022 have been extrapolated for comparability with previous periods. Available data cover the period from 1 January 2022 to 31 May 2022.

**Table 1 healthcare-11-01515-t001:** Demographic characteristics of patients treated within the drug program.

Characteristic	Male	Female	*p*-Value
Patient count	464 (53%)	411 (47%)	
Mean age (range)	21.7 (0–70)	21.3 (0–69)	0.77
Median age	16.2	16.0	
% under 18	53.7%	51.6%	0.61
Patient count by SMA Type:			0.17
SMA 1	117	101	
SMA 2	100	111	
SMA 3	237	189	
presymptomatic	10	10	

## Data Availability

Summary information presented herein is available on demand. Due to the patient data confidentiality policy adopted by the NFZ, no information that would enable single patient identification can be shared. This restriction was also applied during the work on this article.

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
