# Peer review of "Real World Evidence on the Effectiveness of Nusinersen within the National Program to Treat Spinal Muscular Atrophy in Poland"

_healthcare, 2023, doi:10.3390/healthcare11101515_

Round 1

Reviewer 1 Report

The following are the critical issues found:

1) The manuscript is based on the outdated international ICD classification (10 instead of 11).

2) The manuscript reports results that do not innovate the literature and should be improved in its objectives (which cannot simply be to take a snapshot of the factual situation, without elaborating an organizational strategy of the data, detailed limitations, and well-argued future prospects).

3) The list of costs must be justified and detailed so that they can be distinguished and specified.

4) The demographic survey is not supported by a statistical survey of the data. This makes the interest in disseminating these results lower.

5) The conclusions are too schematic, unrepresentative, and without future perspective.

Adequate

Reviewer 2 Report

While the overall topic of the paper may not be the most novel, it seemed that their paper was relatively thorough and made efforts to address any deficiencies in their studies. There are a few minor comments/critiques:t

Line 54: There should a 'The' before the line "First disease modifying agent..."

Lines 99-107: In the methods section, it is a little unclear if 2014 cohort were also included in the drug program, or if there was a way to definitively tell if the 2019 had been treated with nusinersin. It was previously stated that they were studying patients in the drug trial, so this section is confusing in that it is unclear on whether this is the same group or a separate group to that previously mentioned.  If it is a different group, this needs to be better emphasized. If this is the same group, statements about the assumption of treatment and the deaths of patients born in 2014 prior to treatment do not make sense.  

Section 3.1: Sentence on line 138 seems to stop in the middle. Additionally, Figure 2 appears twice. It might be interesting to separate death from ventilation events (if that is presented by one of the two graphs, this should be better described).  

Section 3.2: Is there a difference in the four babies in the drug program that initiated treatment prior to symptoms. Since this is the eventual goal of adding SMA to newborn screening (and may be come the most prevalent demographic of SMA patients in year to come), this might be an interesting note to discuss or an area of further study for the premise of this paper.   Conclusion: The last line of the paper leaves more to be desired. May include a conclusion noting the benefit of this type of analysis to evaluate the cost/benefits of specific treatments or note potential future studies using this data.

Reviewer 3 Report

The manuscript evaluates the effectiveness of nusinersen, a drug used to treat spinal muscular atrophy (SMA), in improving patient care in Poland. The findings suggest that the program is an effective tool for managing the disease and reducing mortality rates, as well as providing a relatively large cohort for analysis. The article's strengths include its use of a national database and the identification of the cost of caring for patients with SMA.

Overall, the article presents valuable insights into the effectiveness of nusinersen in treating SMA and its impact on patient care and healthcare costs in Poland. The study is well-conducted and provides meaningful results that could benefit clinicians and policymakers. Therefore, I recommend this article for publication. However, it would be helpful if the authors could include a in-depth interpretation of the findings and clarify the following issues:

1.Lack of clarity regarding the size and representativeness of the sample population

2. Discuss more on  the heterogeneity of the treated cohort in terms of age and severity. Explain more on how this bias is handled in the experiment.

3.Lack of discussion on the implications of the findings for clinical practice and policy

4. Lack of evaluation of indirect costs related to SMA.

Despite these limitations, the article provides valuable insights into the effectiveness of the nusinersen program in treating SMA patients in Poland. 

Round 2

Reviewer 1 Report

The changes made make the manuscript worthy of publication. The only negative note is the writing style of the references, which needs to be adapted to the journal. Otherwise, it is suitable.